# Clinical Evaluation of Flowable Composite Materials in Permanent Molars Small Class I Restorations: 3-Year Double Blind Clinical Study

**DOI:** 10.3390/ma14154283

**Published:** 2021-07-31

**Authors:** Walter Dukić, Mia Majić, Natalija Prica, Ivan Oreški

**Affiliations:** 1School of Dental Medicine, University of Zagreb, 10000 Zagreb, Croatia; 2Public Health Clinic Daruvar, 43500 Daruvar, Croatia; mia_majic@msn.com; 3Dental Polyclinic Zagreb, 10000 Zagreb, Croatia; natalija_prica@net.hr; 4Clinical Hospital Dubrava, 10000 Zagreb, Croatia; ivan_oreski@yahoo.com

**Keywords:** flowable composite, adhesive technique, class I restorations, clinical performance, clinical, permanent molars

## Abstract

This study evaluated the 3-year clinical performance of four different flowable composite materials used in Small Class I restorations in permanent molars. This double-blinded, clinical study analyzed 229 Small Class I restorations/103 children at baseline, 12, 24, and 36 months with modified United States Public Health Services (USPHS) criteria. The tested flowable materials were Voco Grandio Flow + Voco Solobond M, Vivadent Tetric EvoFlow + Vivadent Excite, Dentsply X-Flow + Dentsply Prime&Bond NT, and 3M ESPE Filtek Supreme XT Flow + 3M ESPE Scotchbond Universal. The retention and marginal adaptation rates were highest for Grandio Flow and X Flow materials after 36 months, resulting in the highest score of clinical acceptability at 95.3% and 97.6%, respectively. The Tetric EvoFlow and Filtek Supreme XT Flow had the same retention rate after 36 months at 88.1%. Statistical significance was found in Grandio flow material in postoperative sensitivity criteria (*p* = 0.021). Tetric EvoFlow showed statistical differences in retention (*p* = 0.01), color match (*p* = 0.004), and marginal adaptation (*p* = 0.042). Filtek Supreme showed statistical differences in retention (*p* = 0.01) and marginal adaptation (*p* < 0.001). The flowable composite materials showed excellent clinical efficacy after 36 months of their clinical usage. There was no difference among the tested flowable composite materials quality in Small Class I restorations over time.

## 1. Introduction

The data indicate that over 80% of dentistry provided in contemporary dental practice is attributed to pit and fissure caries [1,2,3]. Occlusal surfaces represent only 12.5% of the total tooth surface, and 85% of dental caries manifest on occlusal surfaces due to the specific anatomy of molars and the difficulty of adequately removing plaque from them [4]. The minimally invasive caries treatment is based on four modern concepts: Early diagnosis, oral environment modeling based on caries risk evaluation, micro-invasive cavity preparation, dynamic treatment using biologically active materials and modern adhesive systems [5]. Thus, the minimally invasive caries treatment requires removing the bacterial infection and only those dental structures that are irreversibly decayed [6,7]. Moreover, modern resin and adhesive dental materials can prevent tooth structure loss using minimally invasive cavity preparations, enhancing the prognosis of the tooth and becoming the guiding factor in cavity preparation [8,9]. For restoring the tooth, the adhesive technique can be used with a highly filled resin composite material for the prepared pits and fissures, and the remaining pits and fissures can be covered with a sealant [10,11]. The only problem with this type of restoration was that two different materials should be used [12].

Flowable composites have been used since 1995. Previously used flowable composites contained less filler, and as a result of their perceived mechanical limitations, they have traditionally been used clinically for restorations with minimal occlusal loading, such as liners, bases in cavity, Small Class I and II cavities, preventive resin restorations, and Class V lesions [1,12,13,14,15,16,17]. Recently, nanotechnology and the increased filler content have greatly improved the properties of flowable composite materials. Therefore, new types of flowable composite materials were produced for clinical restoration of occlusal cavity with excellent anatomy-forming properties, abrasion resistance, good strength, and expanded clinical application for the occlusal cavity [13,18]. The resulting material is one that flows more easily than traditional composites, making restorations of small preparations easy, especially with an improved delivery system, such as syringes [19,20,21]. For minimally invasive dental restoration procedures, the flowable composite materials are excellent since they preserve a maximum amount of tooth structure and tissue, making them the material of choice in direct posterior restorations for long-term, clinical survival [22]. In addition, flowable materials can be used in minimally invasive preparations and as a sealant material for the non-prepared part of occlusal surface due to their low viscosity [12]. Materials are evaluated through two kinds of tests, in vitro and in vivo, with their advantages and disadvantages [23]. Clinical studies have shown the usage of flowable composites in minimally invasive dentistry regarding Small Class I cavities or preventive resin restorations [8,12,13,23,24,25,26,27,28,29]. Despite the considerable in vitro research regarding flowable composites, there is still insufficient data about their long-term and clinical usage. The aim of this double-blind, clinical study was to evaluate the 3-year performance of four different composite flowable materials used to restore small occlusal caries in permanent molars.

## 2. Materials and Methods

### 2.1. Selection Criteria

For this study, the participants selected included 103 children, aged from 12 to 18 years, with an average age of 15 years and 7 months. Inclusion criteria for patients in the study included the following: (a) Minimum two active occlusal caries lesions on first or second permanent molars confined to occlusal pits and fissures; (b) natural teeth antagonists and contact with opposite tooth; (c) buccolingual width of each restoration not greater that ⅓ the distance between the cusp tips as measured with periodontal probe; (d) a minimum of 1.5 mm in depth; and (e) 20 or more teeth present. The exclusion criteria included (a) severe medical complications; (b) xerostomia; (c) advanced untreated periodontal disease; (e) severe bruxing, clenching or temporomandibular joint disorder; (d) known sensitivity to acrylates or related materials; (e) deep carious defects with frank occlusal cavitation or proximal caries; and (f) poor oral hygiene. Informed consent was obtained from all the individual participants included in the study. The procedures and potential risks, discomforts, and benefits were explained to their parents. An Institutional Review Board and President of Ethics Committee of School of Dental Medicine University of Zagreb approved this study (UPI 034-04/17-6/1; 251-60-4/115-17-3). Informed consent was signed before any dental treatment at the School of Dental Medicine/University Hospital Centre Zagreb (according to HEALTHCARE ACT NN 69-17). The study started in October 2013. The participants were recruited from patients seeking routine dental care at the Department of Pediatric Dentistry. Prior to the clinical procedure, investigators were calibrated using 10 patients with active carious lesions on permanent molars who were not included in this study, and the percentage of agreement between the examiners were at least 85%. The investigators were calibrated for clinical evaluation criteria using photographs for each criteria, and the inter/intra agreement was at least 85%.

### 2.2. Restorative Procedures

Carious lesions were analyzed by visual inspection under standard dental conditions, using standard dental illumination, an air/water from 3 to 1 syringe, and a dental mirror using the visual-ranked method for analysis developed by Ekstrand et al. as a proven technique for analyzing caries, avoiding enamel breakdown using dental explorer in young permanent molars [30]. Prior to the analysis, all the teeth were first prophylactically cleaned with a rotating bristle brush and water to remove the dental plaque and salivary pellicle from occlusal surfaces. Isolation was accomplished using cotton rolls. Patients received local anesthesia prior to the preparation procedure (3% mepivacaine). The restoration procedure removed only carious lesions using small burs using the Komet Micropreparation Kit 4337F (Gebr. Brasseler, Lemgo, Germany) with a high-speed handpiece under constant water cooling. The preparations were made in a way that the buccolingual width was not greater than ⅓ the distance between the cusp tips as measured with a periodontal probe, as mentioned before. The additional “extension for prevention” was avoided and after the lesions were completely excavated, no preparations of undercuts were done. The approximate depth of the cavity was not further than the medium third of the dentine, using visual and tactile feedback from an explorer to determine the end of caries removal [8,10]. Any tooth with a pulp exposure was excluded from the study. All the patients received at least two cavity preparations on permanent molars. The restorative procedures are similar to the work of Yazici and Qin [8,12]. Randomization of materials was performed with sealed opaque envelopes with care to equally distribute the tested materials into tooth type and position variable groups. The patient and the dentist were blinded and unfamiliar about the material used for either tooth. Moreover, the materials were masked with a black tape or permanent marker. Masking materials with a black tape resulted in similar shapes of tested materials, avoiding any favoring of tested materials in clinical application and restorative procedure. Interference in the randomization procedure within patients was performed in order to equally distribute the materials into some important variables such as tooth and position, minimizing the influence of those factors [31].

A unique number code was assigned for each patient for recording the materials used in each tooth. Table 1 shows compositions of the tested materials and the clinical mode of application. Each material was used according to the manufacturer’s instructions. The cavities and fissures were acid etched with phosphoric acid, rinsed, and dried, and the adhesive system was placed and then light cured with Elipar Led Elipar Freelight 2 (3M ESPE, St. Paul, MN, USA), power of 1000 mW/cm^2^, and a wavelength of 430–450 nm. Small increments of flowable composite materials were placed into cavities and light cured to avoid air bubbles, defects, and polymerization shrinkage [28,29].

Abbreviations: Bis-GMA: Bisphenol-glycidine methacrylate; HEMA: Hydroxyethylmethacrylate; TEGDMA: Triethylene glycol dimethacrylate; UDMA: Urethane dimethacrylate; Bis-EMA: Ethoxylated bisphenol-A dimethacrylate; PENTA: Dipentaerythritol penta acrylate monophosphate; HEDMA: Hydroxyethyl dimethacrylate; BHT: Butylated hydroxytoluene; MDP: 10-methacryloyloxydecyl dihydrogen phosphate.

At the end of the treatment, an occlusal check with articulating paper was performed, and an occlusal polishing was accomplished using contouring and finishing diamond burs (Gebr. Brasseler, Lemgo, Germany) at high speed with water cooling. Polishing discs (Sof-Lex; 3M ESPE, St. Paul, MN, USA) and pumice at low speed under water cooling were used for polishing [8]. Restorations were done with cotton rolls isolation, saliva ejectors, and a dental assistant. An experienced dentist performed all the restorative procedures (W.D.). A total of 229 restorations were placed.

### 2.3. Clinical Evaluation Criteria

The restorations were blindly evaluated by another examiner (M.M.) according to the modified criteria from the United States Public Health Services (USPHS) at baseline (1 week), 12, 24, and 36 months [32,33], and only teeth with a complete clinical analysis from baseline to 36 months were included in this study. For each criterion, a score of A indicated the highest degree of clinical acceptability, and B scores indicated clinically acceptable scores, while C and D meant clinically insufficient and unacceptable scores (Table 2). Evaluation was done by the operator using the mirror, dental probe, and 3-in-1 dental syringe from the dental unit. The operator calibration was performed prior to a predetermined level of intra-examiner agreement with at least 85% per each criterion. At the end of the 3-year study, 80 children were available for the analysis.

### 2.4. Statistical Analyses

Statistical analyses were done using SPSS (SPSS Statistics for Windows, v. 17.0. Chicago, IL, USA). Methods of descriptive statistics have been used, and in the case of ordinal variables, nonparametric tests were used. The Kruskal-Wallis test was used to examine the statistical differences between the four materials according to the USPHS criteria (comparison of the same criteria for different materials). The Wilcoxon test of equivalent pairs was used to examine the difference between the results of the evaluation at different time periods with significance of *p* < 0.05. The Friedman non-parametric statistical test was used to examine the statistical differences among different time periods with a significance level of *p* < 0.05. The power analysis of statistical tests was performed using the G*Power v 3.1.3 program (Franz Faul, University of Kiel, Germany). The post-hoc power for tests to detect (d = 0.2) the difference in means between the groups considering a 0.05 alpha error, was calculated to be 0.8, with a total sample size of 168.

## 3. Results

The recall rates are shown in Table 3. After 3 years, the loss of samples was 29.5% for GF, 28.1% for XF, 25% for FS, and 23.9% for TF. The Chi-square test showed no statistical significance in the number of recall rates among the tested materials for different time periods, in each material and in total count (*p* > 0.05). Figure 1 shows the CONSORT flow diagram (Checklist is in Appendix A).

The results of the clinical evaluation of four tested materials for baseline, 12, 24, and 36 months are shown in Table 4. None of the tested materials resulted in a score of D in any criteria, not for 12, 24 or 36 months.

The retention rates were highest for XF and GF materials after 36 months, resulting in a score of A at 97.6% and 95.3%, respectively. Marginal adaptation rates were also highest after 36 months for XF and TF, resulting in 97.6% and 95.2%, respectively. The TF and FS had the same retention rate after 36 months at 88.1%. The color match showed a change in the period of 24 and 36 months, resulting in a score of C in 2.3% for GF, 9.5% for TF, and 2.4% for FS materials. The marginal discoloration score of C was found in 2.3% of cases only in the GF material after 36 months. The surface texture showed a score of B in the GF (2.3%) and X-Flow (2.4%) materials after 36 months. The anatomic form showed only a score of A after 24 and 36 months in all the tested materials. No secondary caries was detected between any tested materials after 12, 24 or 36 months. A score of B was found for postoperative sensitivity in TF and XF materials after 36 months in 7.1% and 4.9%, respectively. The Friedman test showed statistical differences in postoperative sensitivity criteria for the GF material (*p* = 0.021). TF showed statistical differences in retention (*p* = 0.01), color match (*p* = 0.004), and marginal adaptation (*p* = 0.042). FS showed statistical differences in retention (*p* = 0.01) and marginal adaptation (*p* < 0.001). Table 5 shows the results of tested materials considering the USPHS criteria and location in maxillary teeth or mandibular teeth. Considering *p* < 0.05, there was no statistical significance among the tested materials regarding the location in maxillary or mandibular region. The analysis of difference between the 1st or 2nd molar and the tested materials through a period baseline of 36 months showed only statistical differences in marginal adaptation criteria (*p* = 0.033) and the results are in Table 6.

## 4. Discussion

In this study, four different flowable materials in Small Class I restorations for the periods of 12, 24, and 36 months have been analyzed. After the period of 36 months, all the restorations showed good clinical results, with no D scores among the four flowable composite materials.

Qin stated that flowable composite materials were not suggested for use as restorations in occlusal molar caries due to their resistance and fracture. Moreover, abrasion and pressure resistance were not important factors compared to retention and marginal adaptation since the restorations were in narrow cavities that were very small, within 1.5 mm width [12]. According to Simonsen, it is important to perform only conservative caries removal using the smallest and precise burs, restore the cavity using the acid-etch technique and the contemporary restorative resin composite material, as was used here [34]. Strassler and Goodman introduced the use of a flowable composite material for the preventive resin restoration technique and provided a 5-year follow-up study with good clinical results [35]. In addition, Qin et al. showed 96.55% of flowable resin composites and 93% of flowable compomers in cavities still complete with no definition of caries present, compared to a completion of 86.5% for conventional composite in cavities. The authors suggested that flowable resin composites and flowable compomers can be used for preventive resin restorations [12]. Sabbagh et al. analyzed the new self-adhering flowable composite Vertise flow and the conventional Premise Flowable in Small Class I restorations after a 2-year clinical period. The two resin-based materials showed similar retention rates, but the conventional flowable composite with an adhesive system showed a better retention rate [23]. Shaalan analyzed self-adhering flowable composite versus flowable composite in conservative Class I cavities after 6 months, resulting in no statistically significant differences between both materials for all the tested outcomes [24]. Braem also analyzed the self-adhering flowable and conventional flowable resin composite, resulting in a similar clinical performance at the 5-year follow-up period. Both materials showed some degradation over time regarding marginal adaptation and marginal discoloration [25]. Another clinical study analyzed the quality of minimally invasive occlusal restorations restored with the Grandio Flow nano-filled composite, according to the two different cavity preparation methods. No difference was observed between the two methods of cavity preparations in terms of marginal adaptation and discoloration. Retention rates after 2 years were 98.1% for bur and 100% for the laser group. The retention rate score of A for the Grandio Flow in the bur group after 24 months was 98.1%, and it was similar to our results of 96%. In addition, the other clinical parameters showed a score of A in 90% of cases after the 2-year clinical period. The authors suggested using lasers in minimally invasive resin composite cavity preparations, but long-term recalls are planned to determine whether differences in clinical performance between the two methods of cavity preparations will occur at later restoration ages [8]. Another study analyzed a clinical evaluation after 3 years of Tetric flow and X Flow in Class I restorations. Marginal adaptation after 3 years in the A group had dropped to 51.7% for Tetric EvoFlow and 65.5% for X-Flow. Only the color matching exhibited an overall difference between the treatment groups, but a clear majority of the restorations were still successfully functioning and acceptable after 36 months, which is in accordance with our results. The authors suggest using flowable composites for restorations of ¼ or less the width of distance between the cusp tips [29]. Kitasako et al. found that the highly filled flowable composite in posterior restorations is not superior to the conventional composite restorations after 36 months [36]. Lawson et al. found that the flowable composite Filtek Supreme Ultra and conventional composite Filtek Supreme Ultra Universal have similar properties after 24 months in conservative Class I restorations with an isthmus less than ½ the intercuspal distance [13]. Marginal adaptation for Filtek Supreme Ultra flowable was 85.7% after 24 months, and it was similar to our findings of 83.3%. In addition, there was a slight change in color match in 85.7% after 24 months, in accordance with our data of 81%. The authors reported secondary caries in 5.5% of cases after 24 months, but we did not report any caries in our study. Similar clinical studies have shown also zero-cell count regarding retention, anatomic form, surface texture, postoperative sensitivity, caries or color match. There is a difference in some clinical variables through time, resulting in a variety of materials of clinical performance [8,25,28,29,36]. In our study, statistical differences were found in postoperative sensitivity in GF material. Retention, color match, and marginal discoloration were statistically significant in TF material. FS showed statistical significance in retention and marginal adaptation. We can conclude from previous clinical studies and our findings that flowable composite materials report good clinical results when used in Small Class I restorations. The recall rates after 3 years were from 70.5% to 76.9% and it is similar to our previous study [25]. In our study, we use flowable composite materials, which are restoration materials for Small Class I restorations, although new bulk fill flowable materials were introduced recently. All the tested materials showed good clinical results after 36 months, and there was no secondary caries in any of the cases. Additionally, slight differences were found among the flowable tested materials, which can be attributed to differences in material preparation, chemical composition, and adhesives used. Many reports support the use of flowable composites in minimally invasive dentistry. Among the dental practitioners in Germany, 78.6% of them use a flowable composite for posterior restorations, and 74.2% use them for Small Class I restorations. Regarding these data, it might be concluded that, in addition to the different handling, the leading motivation for dentists to use flowable composites is the increase in the quality of their restorations with respect to time savings [14]. The tested materials were applied with different adhesives, which can affect the final outcome in the clinical analysis of flowable materials. Another study showed that 72% of pediatric dentists perform preventive resin restorations in their offices, most commonly restored with the flowable composite resin [37]. Flowable composites have become a broadband material in many aesthetic dental procedures and are considered as a promising material for the future [17]. The dental literature favors the use of highly filled resin composites in restoration of small pit and fissure caries where conservative preventive resin restorations are indicated in both the primary and permanent dentition [1]. In contrast, the selection of materials for restorations should be chosen by the location and size of the lesion, the caries risk, lesion activity, particular patient conditions, and environment. The authors conclude that there is lack of evidence in choosing the right materials for tooth restorations after selective carious tissue removal to soft or firm dentine [15,38]. Minimally invasive restorations represent scientifically documented advantages over extensive and tissue-destructive traditional restorations. Therefore, they preserve the strength of the residual tooth structure using optimal adhesive restoratives [15,39]. The meta-analysis from a recent study concluded that there was no statistical or clinical difference between flowable and conventional composites in posterior restorations. New generations of flowable composites were developed using nanotechnology creating a composite with properties of conventional resin composites. This enhanced the mechanical and physical properties of flowable composites and enabled their usage in Small Class I restorations [40]. We suggest, based on many previous clinical scientific reports and results from our study, that the material of choice in Small Class I restorations is the flowable composite, with excellent clinical properties, clinical handling, longevity, and durability. Flowable composites are the recommended material of choice for the Small Class I restorations and ultra-conservative restorations. All four tested flowable composite materials showed similar clinical results. However, long-term recalls and future studies should be performed to determine the clinical differences among new flowable composite materials on the dental market, regarding new bulk-fill flowable composites. The low, medium or high risk of caries among young patients should be considered as an important variable in futures studies, which could have an impact on restorations quality and longevity. In addition, the limitations of this study included the limited number of tested flowable composite materials, only urban children population, unknown dietary, oral hygiene habits, and different methodologies (some teeth were tested within the same subjects), which can all influence the final results.

## 5. Conclusions

The flowable composite materials showed excellent clinical efficacy after 36 months of their clinical usage. At 3 years, the tested flowable composite material quality in Small Class I restorations showed similar clinical results. This study suggested flowable composites and the adhesive technique as the material of choice used in Small Class I restorations. Future clinical studies should compare the materials such as bulk fill or self-adhering flowable composites.

## Figures and Tables

**Figure 1 materials-14-04283-f001:**
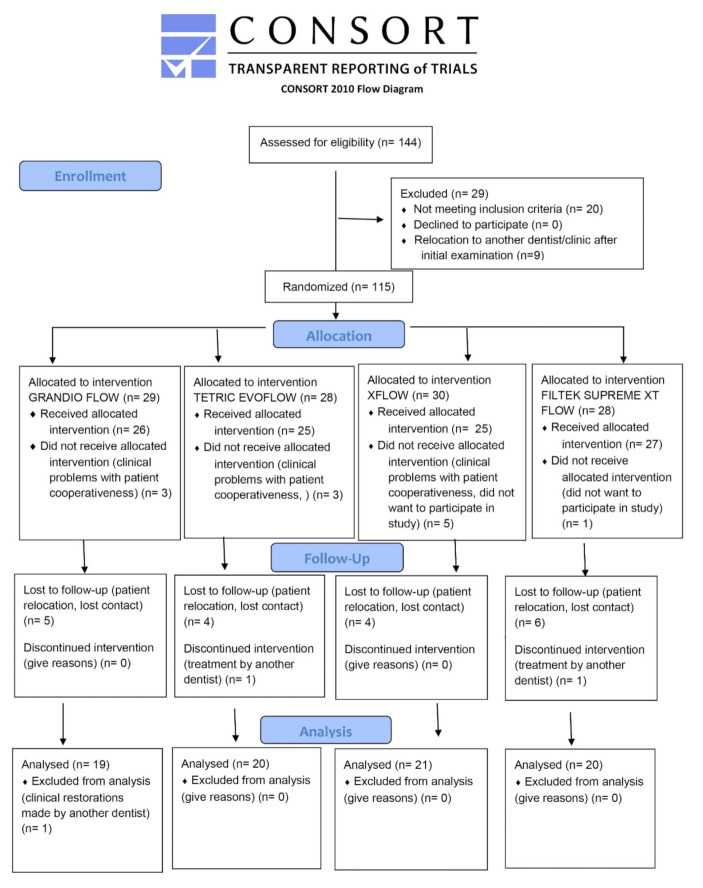
The CONSORT flow diagram.

**Table 1 materials-14-04283-t001:** Compositions of tested materials and the clinical mode of application.

Flowable Material/Adhesive System/GROUPS	Composition	Mode of Clinical Application
Grandio Flow (Voco, Cuxhaven, Germany) Voco Solobond M (Voco, Cuxhaven, Germany) GROUP GF	Inorganic glass ceramic fillers, Bis-GMA, TEGDMA, HEDMA, inorganic filler 80% loading by weight. Bis-GMA, HEMA, BHT, acetone organic acids.	Acid etching 34.5% phosphoric acid (30 s enamel, 15 s dentine), rinsing (20 s), gently air drying (2 s) of dentine leaving it moist, adhesive application Solobond, light blowing (2 s), light curing (20 s), application of resin composite, light cure (40 s).
Tetric Evoflow (Vivadent, Schaan, Liechtenstein) Excite (Vivadent, Schaan, Liechtenstein GROUP TF	Bis-GMA, UDMA, Barium glass filler, Ytterbiumtrifluoride, Mixed oxide, Highly dispersed silica, 62% inorganic filler loading by weight. Phosphonic acid acrylate, HEMA, Bis-GMA Dimethacrylate, Highly dispersed silicon dioxide, Ethanol.	Acid etching 37% phosphoric acid (30 s enamel, 15 s dentine), rinsing (20 s), gently air drying of dentine (2 s) leaving it moist, adhesive application Excite, light blowing (2 s), light curing (20 s), application of resin composite, light cure (40 s).
Xflow (Dentsply Sirona, York, PA, USA) Prime&Bond NT (Dentsply Sirona, York, PA, USA) GROUP XF	Urethane modified BisGMA-adduct, Bis-GMA and diluents, nanofiller silica, 62% inorganic filler loading by weight. PENTA, UDMA, Resin R5-62-1, T-resin, D-resin, nanofiller, acetone, and cetylaminehydrofluoride.	Acid etching 37% phosphoric acid (30 s enamel, 15 s dentine), rinsing (20 s), gently air drying of dentine (2 s) leaving it moist adhesive application Prime&Bond NT, light blowing (2 s), light curing (20 s), application of resin composite, light cure (40 s).
Filtek Supreme XT Flow (3M/Espe, St. Paul, MN, USA) Scotchbond Universal (3M/Espe, St. Paul, MN, USA) GROUP FS	Bis-GMA, TEGDMA, and Bis-EMA, dimethacrylate polymer, silica and zirconia nanofiller, 65% inorganic filler loading by weight. MDP Phosphate Monomer, Dimethacrylate resins, HEMA, silane, Ethanol.	Acid etching 37% phosphoric acid (30 s enamel, 15 s dentine), rinsing (20 s), gently air drying of dentine (2 s) leaving it moist, adhesive application Scotchbond Universal, light blowing (2 s), light curing (20 s), application of resin composite, light cure (40 s).

**Table 2 materials-14-04283-t002:** Modified criteria from the United States Public Health Services. A: Alpha score. B: Bravo score. C: Charlie score. D: Delta score.

Score/ Criteria	Retention	Color Match	Marginal Discoloration	Marginal Adaptation	Anatomic Form	Surface Texture	Postoperative Sensitivity	Secondary Caries
A	restoration is present	Excellent match of color and translucency compared to the neighboring tooth tissue, restoration almost invisible	No discoloration	No visible gap or crevice, probe does not catch or penetrate	The restoration is continuous with tooth anatomy, ideal	restoration surface is smooth as the surrounding enamel	no postoperative sensitivity	no caries is present at the margin of the restoration, as evidenced by softness and opacity
B	partial loss of retention	Slight mismatch, only visible by close examination	Minor staining, can be polished	Visible gap or crevice, slight catching or penetration of probe	Slightly under or over contoured restoration, no dentin exposed	surface rougher than the surrounding enamel	Slight and mild occasional sensitivity	
C	restoration absent	Moderate mismatch in color, shade or translucency	Moderate surface staining, not aesthetically unacceptable	Visible gap or extensive probe penetration between cavity wall and restoration		Restoration is under contoured, dentin or base exposed, restorative material missing, failure		
D		Extensive color mismatch, outside the limits of acceptable appearance	Surface staining present on the restoration, intervention necessary	Loose restoration, secondary caries			constant sensitivity	caries present

**Table 3 materials-14-04283-t003:** Recall rates.

Material	Baseline	12 Months	24 Months	36 Months	*p* *	*p* **
GF	61(100%)	57(93.4%)	50 (81.9%)	43 (70.5%)	0.996	0.999
TF	55 (100%)	53 (96.3%)	47 (85.5%)	42 (76.3%)	0.996	
XF	57 (100%)	53 (92.5%)	48 (84.2%)	41 (71.9%)	1.000	
FS	56 (100%)	53 (94.6%)	48 (85.7%)	42 (75%)	0.999	
Total	229 (100%)	216 (94.3%)	193 (84.2%)	168 (73.3%)		

*, ** Chi-square test statistical significance.

**Table 4 materials-14-04283-t004:** Analysis of tested materials through the period of baseline, 12, 24, and 36 months considering USPHS criteria. A: Alpha score. B: Bravo score. C: Charlie score. D: Delta score. * Statistical significance.

	Measurement									*p* *
	Baseline		12 Months		24 Months		36 Months			
	A	B	A	B	A	B	A	B	C	
GF	*n* (%)		*n* (%)		*n* (%)		*n* (%)			
Retention	43 (100.0)	0 (0.0)	42 (97.7)	1 (2.3)	42 (97.7)	1 (2.3)	41 (95.3)	1 (2.3)	1 (2.3)	0.750
Color match	35 (81.4)	8 (18.6)	35 (81.4)	8 (18.6)	35 (81.4)	8 (18.6)	34 (79.1)	8 (18.6)	1 (2.3)	1.000
Marginal discoloration	41 (95.3)	2 (4.7)	41 (95.3)	2 (4.7)	41 (95.3)	2 (4.7)	40 (93.0)	2 (4.7)	1 (2.3)	1.000
Marginal adaptation	43 (100.0)	0 (0.0)	41 (95.3)	2 (4.7)	41 (95.3)	2 (4.7)	40 (93.0)	3 (7.0)	0 (0.0)	0.188
Caries	43 (100.0)	0 (0.0)	43 (100.0)	0 (0.0)	43 (100.0)	0 (0.0)	43 (100.0)	0 (0.0)	0 (0.0)	1.000
Surface texture	43 (100.0)	0 (0.0)	43 (100.0)	0 (0.0)	43 (100.0)	0 (0.0)	42 (97.7)	1 (2.3)	0 (0.0)	1.000
Anatomic form	43 (100.0)	0 (0.0)	43 (100.0)	0 (0.0)	43 (100.0)	0 (0.0)	43 (100.0)	0 (0.0)	0 (0.0)	1.000
Postoperative sensitivity	39 (90.7)	4 (9.3)	41 (95.3)	2 (4.7)	43 (100.0)	0 (0.0)	43 (100.0)	0 (0.0)	0 (0.0)	0.021 *
TF	*n* (%)		*n* (%)		*n* (%)		*n* (%)			
Retention	42 (100.0)	0 (0.0)	41 (97.6)	1 (2.4)	39 (92.9)	3 (7.1)	37 (88.1)	4 (9.5)	1 (2.4)	0.010 *
Color match	36 (85.7)	6 (14.3)	36 (85.7)	6 (14.3)	36 (85.7)	6 (14.3)	32 (76.2)	6 (14.3)	4 (9.5)	0.004 *
Marginal discoloration	42 (100.0)	0 (0.0)	41 (97.6)	1 (2.4)	39 (92.9)	3 (7.1)	38 (90.5)	4 (9.5)	0 (0.0)	0.042 *
Marginal adaptation	42 (100.0)	0 (0.0)	42 (100.0)	0 (0.0)	41 (97.6)	1 (2.4)	40 (95.2)	2 (4.8)	0 (0.0)	0.500
Caries	42 (100.0)	0 (0.0)	42 (100.0)	0 (0.0)	42 (100.0)	0 (0.0)	42 (100.0)	0 (0.0)	0 (0.0)	1.000
Surface texture	42 (100.0)	0 (0.0)	42 (100.0)	0 (0.0)	42 (100.0)	0 (0.0)	42 (100.0)	0 (0.0)	0 (0.0)	1.000
Anatomic form	42 (100.0)	0 (0.0)	42 (100.0)	0 (0.0)	42 (100.0)	0 (0.0)	42 (100.0)	0 (0.0)	0 (0.0)	1.000
Postoperative sensitivity	39 (92.9)	3 (7.1)	40 (95.2)	2 (4.8)	39 (92.9)	3 (7.1)	39 (92.9)	3 (7.1)	0 (0.0)	1.000
XF	*n* (%)		*n* (%)		*n* (%)		*n* (%)			
Retention	41 (100.0)	0 (0.0)	41 (100.0)	0 (0.0)	41 (100.0)	0 (0.0)	40 (97.6)	1 (2.4)	0 (0.0)	1.000
Color match	36 (87.8)	5 (12.2)	36 (87.8)	5 (12.2)	36 (87.8)	5 (12.2)	36 (87.8)	5 (12.2)	0 (0.0)	1.000
Marginal discoloration	41 (100.0)	0 (0.0)	41 (100.0)	0 (0.0)	39 (95.1)	2 (4.9)	39 (95.1)	2 (4.9)	0 (0.0)	0.167
Marginal adaptation	41 (100.0)	0 (0.0)	41 (100.0)	0 (0.0)	41 (100.0)	0 (0.0)	40 (97.6)	1 (2.4)	0 (0.0)	1.000
Caries	41 (100.0)	0 (0.0)	41 (100.0)	0 (0.0)	41 (100.0)	0 (0.0)	41 (100.0)	0 (0.0)	0 (0.0)	1.000
Surface texture	41 (100.0)	0 (0.0)	41 (100.0)	0 (0.0)	41 (100.0)	0 (0.0)	40 (97.6)	1 (2.4)	0 (0.0)	1.000
Anatomic form	41 (100.0)	0 (0.0)	41 (100.0)	0 (0.0)	41 (100.0)	0 (0.0)	41 (100.0)	0 (0.0)	0 (0.0)	1.000
Postoperative sensitivity	39 (95.1)	2 (4.9)	39 (95.1)	2 (4.9)	39 (95.1)	2 (4.9)	39 (95.1)	2 (4.9)	0 (0.0)	1.000
FS	*n* (%)		*n* (%)		*n* (%)		*n* (%)			
Retention	42 (100.0)	0 (0.0)	40 (95.2)	2 (4.8)	38 (90.5)	4 (9.5)	37 (88.1)	4 (9.5)	1 (2.4)	0.010 *
Color match	35 (83.3)	7 (16.7)	35 (83.3)	7 (16.7)	35 (83.3)	7 (16.7)	34 (81.0)	7 (16.7)	1 (2.4)	1.000
Marginal discoloration	40 (95.2)	2 (4.8)	40 (95.2)	2 (4.8)	40 (95.2)	2 (4.8)	39 (92.9)	3 (7.1)	0 (0.0)	1.000
Marginal adaptation	42 (100.0)	0 (0.0)	36 (85.7)	6 (14.3)	35 (83.3)	7 (16.7)	35 (83.3)	7 (16.7)	0 (0.0)	<0.001 *
Caries	42 (100.0)	0 (0.0)	42 (100.0)	0 (0.0)	42 (100.0)	0 (0.0)	42 (100.0)	0 (0.0)	0 (0.0)	1.000
Surface texture	42 (100.0)	0 (0.0)	42 (100.0)	0 (0.0)	42 (100.0)	0 (0.0)	42 (100.0)	0 (0.0)	0 (0.0)	1.000
Anatomic form	42 (100.0)	0 (0.0)	42 (100.0)	0 (0.0)	42 (100.0)	0 (0.0)	42 (100.0)	0 (0.0)	0 (0.0)	1.000
Postoperative sensitivity	39 (92.9)	3 (7.1)	40 (95.2)	2 (4.8)	42 (100.0)	0 (0.0)	42 (100.0)	0 (0.0)	0 (0.0)	0.083

**Table 5 materials-14-04283-t005:** Analysis of tested materials considering USPHS criteria and location in maxilla or mandibula jaw.

Material		GF		TF		XF		FS	
	Tooth Location	Mean Rank	*p* *	Mean Rank	*p* *	Mean Rank	*p* *	Mean Rank	*p* *
Retention: 36 months—Baseline	maxilla	21.85	0.785	23.04	0.065	21.39	0.376	22.62	0.386
	mandible	22.24		19.00		20.50		20.74	
Color Match: 36 months—Baseline	maxilla	22.33	0.419	20.60	0.279	21.00	1.000	21.00	0.410
	mandible	21.50		22.97		21.00		21.84	
Marginal discoloration: 36 months—Baseline	maxilla	22.33	0.419	22.73	0.103	20.89	0.860	21.00	0.410
	mandible	21.50		19.50		21.14		21.84	
Marginal adaptation: 36 months—Baseline	maxilla	22.15	0.822	22.12	0.261	20.50	0.258	20.47	0.487
	mandible	21.76		20.50		21.64		22.20	
Caries: 36 months—baseline	maxilla	22.00	1.000	21.50	1.000	21.00	1.000	21.50	1.000
	mandible	22.00		21.50		21.00		21.50	
Surface texture: 36 months—Baseline	maxilla	22.33	0.419	21.50	1.000	20.50	0.258	21.50	1.000
	mandible	21.50		21.50		21.64		21.50	
Anatomic form: 36 months—Baseline	maxilla	22.00	1.000	21.50	1.000	21.00	1.000	21.50	1.000
	mandible	22.00		21.50		21.00		21.50	
Postoperative sensitivity: 36 months—Baseline	maxilla	21.52	0.537	21.50	1.000	21.00	1.000	20.53	0.343
	mandible	22.74		21.50		21.00		22.16	

* Statistical significance.

**Table 6 materials-14-04283-t006:** Analysis of tested materials considering USPHS criteria and 1st or 2nd molar.

Material		GF		TF		XF		FS	
	Tooth	Mean Rank	*p* *	Mean Rank	*p* *	Mean Rank	*p* *	Mean Rank	*p* *
Retention: 36 months—Baseline	1st molar	22.00	1.000	22.25	0.369	21.26	0.471	22.13	0.427
	2nd molar	22.00		20.28		20.50		20.37	
Color match: 36 months—Baseline	1st molar	22.48	0.329	21.35	0.854	21.00	1.000	21.78	0.456
	2nd molar	21.50		21.75		21.00		21.00	
Marginal discoloration: 36 months—Baseline	1st molar	21.50	0.306	21.12	0.611	21.52	0.302	21.00	0.180
	2nd molar	22.52		22.13		20.00		22.40	
Marginal adaptation: 36 months—Baseline	1st molar	23.43	0.083	21.31	0.726	21.26	0.471	19.56	0.033 *
	2nd molar	20.50		21.81		20.50		25.00	
Caries: 36 months—Baseline	1st molar	22.00	1.000	21.50	1.000	21.00	1.000	21.50	1.000
	2nd molar	22.00		21.50		21.00		21.50	
Surface texture: 36 months—Baseline	1st molar	21.50	0.306	21.50	1.000	21.26	0.471	21.50	1.000
	2nd molar	22.52		21.50		20.50		21.50	
Anatomic form: 36 months—Baseline	1st molar	22.00	1.000	21.50	1.000	21.00	1.000	21.50	1.000
	2nd molar	22.00		21.50		21.00		21.50	
Postoperative sensitivity: 36 months—Baseline	1st molar	21.07	0.322	21.50	1.000	21.00	1.000	20.67	0.186
	2nd molar	22.98		21.50		21.00		23.00	

* Statistical significance.

## Data Availability

The data presented in this study are available on request from the corresponding author.

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
