# Peer review of "Clinical Evaluation of Flowable Composite Materials in Permanent Molars Small Class I Restorations: 3-Year Double Blind Clinical Study"

_materials, 2021, doi:10.3390/ma14154283_

Round 1

Reviewer 1 Report

Dental caries is one of the most prevalent diseases worldwide. Restoration of endodontically treated teeth has always been a challenging topic for dentists, as complications may ultimately result in tooth loss if the correct restorative decision is not made. There are numerous choices of restorative materials and restorations, with limited guidance on the best approaches in different circumstances. Therefore, comparative clinical studies of existing new drugs are necessary and relevant. The flowable composite materials showed in vitro high quality and needed mechanic property and clinical efficacy. The article is written in clear language. However, there are some comments.

  1. The quality of the graphs, part of the figure is cut off (Fig. 2)
  2. Figure 1 does not show the statistical difference between groups.
  3. No test group using generally accepted Resin composites material

Author Response

  1. The quality of the graphs, part of the figure is cut off (Fig. 2)

The graph, Figure 2 has been replaced with corrected figure.

  1. Figure 1 does not show the statistical difference between groups.

We did not analyzed the statistical significance among loss of participants because we performed power analysis post hoc, resulting in total number of 168 restorations statistically adequate for study. This part is now inserted into Statistical analysis part. Quote: Power analysis of statistical tests was performed using G*Power v 3.1.3 program (Franz Faul, University of Kiel, Germany). Power of tests post hoc to detect (d=0.2) difference in means between groups considering a 0.05 alpha error, was calculated to be 0.8, with total sample size of 168.

  1. No test group using generally accepted Resin composites material.

We used similar clinical protocol as published before and authors in their clinical study did not had control group/placebo among tested materials because they choose to perform active control study. Their clinical study and our is active control study, which is more ethical than placebo/control group, eliminating risks for patients and resulting that children/patients are all treated with quality materials for restorations. Moreover, the main goal is analysis of materials quality after 3 years of clinical usage, because small class I restorations with modern composite materials are still new approach to some dentists and results from this study can improve their clinical performance and knowledge.

 (Qin M, Liu H. Clinical evaluation of a flowable resin composite and flowable compomer for preventive resin restora-tions. Oper Dent 2005. Yazici AR, Baseren M, Gorucu J. Clinical comparison of bur- and laser prepared minimally invasive occlusal resin composite restorations: two-year follow-up. Oper Dent 2010. Lawson NC, Radhakrishnan R, Givan DA et al. Two-year Randomized, Controlled Clinical Trial of a Flowable and Conventional Composite in Class I Restorations. Oper Dent 2015.  Sabbagh J, Dagher S, El Osta N et al. Randomized Clinical Trial of a Self-Adhering Flowable Composite for Class I Restorations: 2-Year Results. Int J Dent 2017,2017:5041529.)

Reviewer 2 Report

Dear Authors

This ms is an interesting randomized clinical trial about the clinical performance of four different flowable composite materials used in small Class I restorations in permanent molars. Although the study is interesting, details are missing in the Materials and Methods section. For this reason, I suggest a minor revision to resolve this, and other points described as following :

Abstract

  1. Have the authors considered the use of some abbreviation for the groups? It might improve the readability of the text.
  2. The authors should better specify the results. For example, Line 19 “Statistical significance was found in Grandio flow material”. About what the authors found statistical significance? Retention rate? Marginal adaptation? Or both.

Introduction

  1. Line 49: In which way have surface preparation technology improved the flowable materials? The authors should better specify this point.
  2. Lines 66: I suggest using only one term to identify your work: double blind clinical study (wrote in the abstract) or randomized study.

Materials and Methods

  1. I suggest moving Lines 78-89 at the beginning of the paragraph.
  2. The authors might specify in the title that the study was performed in young patient.
  3. Lines 95-97: the authors should specify the reason in not using tactile method.
  4. How have the authors calculated the number of teeth?
  5. The authors should specify the method for randomization.
  6. Lines 134-135: Separate the figure legend with the main text.
  7. Was occlusal check performed at the end of the treatment?
  8. Table 1: Correct the space in the Tetric Evoflow section in pag. 4
  9. The authors should better explain the reason in not using rubber dam for isolation.
  10. Lines 147-151: Did WD participate during the clinical evaluation? How many operators were involved?
  11. The authors should specify if they also evaluated the depth of the I Class Cavity, since the distance from the light curing tip is detrimental for the correct polymerization of resin materials.

Discussion

  1. Line 258: “Is seems that some variables 258 for tested materials shows less degradation problems through time” rephrase this sentence.
  2. I strongly suggest changing the title accordingly with the main text. For example, “Clinical evaluation of flowable composite materials in permanent molars small Class I restorations in young patient: 3-year randomized trial”
  3. Lines 301-304: the authors might consider also the important of patient caries susceptibility.

Conclusion

  1. The authors should explain the conclusion following the author guidelines and other articles in Materials Journal. I recommend to not use pointed style in the conclusion section.

Author Response

Abstract

  1. Have the authors considered the use of some abbreviation for the groups? It might improve the readability of the text.

Corrected.

  1. The authors should better specify the results. For example, Line 19 “Statistical significance was found in Grandio flow material”. About what the authors found statistical significance? Retention rate? Marginal adaptation? Or both.
    Corrected in the text.

Introduction

  1. Line 49: In which way have surface preparation technology improved the flowable materials? The authors should better specify this point.
    Corrected.
  2. Lines 66: I suggest using only one term to identify your work: double blind clinical study (wrote in the abstract) or randomized study.
    Corrected.

Materials and Methods

  1. I suggest moving Lines 78-89 at the beginning of the paragraph.
    Corrected.
  2. The authors might specify in the title that the study was performed in young patient.
    We have changed the title as requested into “Clinical evaluation of flowable composite materials in permanent molars small Class I restorations: 3-year double blind clinical study”. If we add “young patient” if will be too long and Journal will be shorted. If reviewers insist, we can use suggested title.
  3. Lines 95-97: the authors should specify the reason in not using tactile method.
    This method Ekstrand 1997 was chosen because it excludes dental probe and force, avoiding hard tissue damage and possible enamel breakdown by explorer’s force which is very common in young permanent molars. Most of published papers in our data analysis prior to this clinical trial has chosen Ekstrand 1997 method for analysis, so we did. Additional explanation was inserted into text.
  4. How have the authors calculated the number of teeth?

The number of teeth for clinical trial has been calculated and analyzed with statistical software (Franz Faul, University of Kiel, Germany).

  1. The two versions have been calculated, a priori and post hoc. In both cases were adequate number of teeth for statistical analysis, in a priori test were 164 teeth (d=0,2, α err prob 0,05 i power 0,8) and in post hoc test were 168 teeth (d=0,2, α err prob 0,05 i power 0,8). Both analyses satisfy, and we  chosen post hoc analysis. It is inserted into text.
  1. The authors should specify the method for randomization.
    Randomization of materials was performed with sealed opaque envelopes with care to equally distribute the tested materials into tooth type and position variable groups. Similar methods have been described in previously published papers.
  2. Lines 134-135: Separate the figure legend with the main text.

    Corrected.
  3. Was occlusal check performed at the end of the treatment?

Corrected.

  1. Table 1: Correct the space in the Tetric Evoflow section in pag. 4

Corrected.

  1. The authors should better explain the reason in not using rubber dam for isolation.
    There are papers which were using rubber dam, and papers without rubber dam. Before we started this trial, in our meta analysis of previously published papers and data searc, resulted that most of them did not use rubber dam( Qin-Operative Dentistry 2005, Yazici-Operative Dentistry 2010, Demirici-J Adhes Dent 2007, Demirici –Am J  Dent 2006,Gallo Quintessence 2006, Gallo Quintessence 2010, Lawson-Operative Dentistry 2015,Cehreli J of Dentistry 2000) and only two used rubber dam, one published 2017 (Dresch Operativre Dentistry 2005, Sabbagh Int J Dentistry 2017). The average age of our patients were 15years, and we were able to have dry operational field with cotton rolls and saliva ejectors, as described in previously published clinical papers. There is possibility to use rubber dam, but it could make clinical procedure more difficult because of possible fear of rubber dam instruments for young patient, prolonged time for adapting rubber dam and additional usage of surgical sauger. We suggest to use rubber dam in adult patients or in special cases where dry operational field is not possible.
  2. Lines 147-151: Did WD participate during the clinical evaluation? How many operators were involved?
    WD was not included into clinical examination/evaluation, MM was in charge for clinical examination. It is explained in the study.
  1. The authors should specify if they also evaluated the depth of the I Class Cavity, since the distance from the light curing tip is detrimental for the correct polymerization of resin materials.
    The approximate depth of the cavity was not further than the medium third of the dentine, as described before in papers (Yazici AR, Baseren M, Gorucu J. Clinical comparison of bur- and laser prepared minimally invasive occlusal resin composite restorations: two-year follow-up. Oper Dent 2010,35,500-507. Simonsen RJ. Preventive resin restorations (I). Quintessence Int Dent Dig 1978,9,69-76.) The cavities were not deeper than medium third of dentine, and curing lights tip were perpendicular to occlusal surface and restoration, so the polymerization was according to manufactures recommendations. The LED lamp has power of 1000 mW/cm2, and we used composite is small increments, resulting in less polymerization shrinkage and correct polymerization of materials.

Discussion

  1. Line 258: “Is seems that some variables 258 for tested materials shows less degradation problems through time” rephrase this sentence.
    Corrected.
  2. I strongly suggest changing the title accordingly with the main text. For example, “Clinical evaluation of flowable composite materials in permanent molars small Class I restorations in young patient: 3-year randomized trial”

Corrected. New title and corrected in the text.

  1. Lines 301-304: the authors might consider also the important of patient caries susceptibility.
    Corrected and inserted into text.

Conclusion

  1. The authors should explain the conclusion following the author guidelines and other articles in Materials Journal. I recommend to not use pointed style in the conclusion section.

Corrected.

Round 2

Reviewer 1 Report

Still, Figure 1 presents the results of the study group analysis. Variations (standard deviation, error of the mean, etc., depending on the normal distribution of the data) should be reflected

Author Response

Dear Reviewer, we erased Figure 1 and inserted new Table 3 with statistical analysis among time periods and materials groups, as recommended by our statistician. The chi square test showed  no statistical significance among time periods (baseline,12m, 24m,36m) for each material and overall. There is no statistical difference in number of teeth at baseline and after 36 months (p>0.05). The  clinical drop-out in our study, is similar to other clinical published papers.